# Histone Methylation and Memory of Environmental Stress

**DOI:** 10.3390/cells8040339

**Published:** 2019-04-10

**Authors:** Paola Fabrizio, Steven Garvis, Francesca Palladino

**Affiliations:** Laboratory of Biology and Modeling of the Cell, Ecole Normale Supérieure de Lyon, CNRS, Université Claude Bernard Lyon 1, 69007 Lyon, France; paola.fabrizio@ens-lyon.fr (P.F.), steve.garvis@ens-lyon.fr (S.G.)

**Keywords:** chromatin, histone methylation, transcriptional memory, epigenetic inheritance, stress

## Abstract

Cellular adaptation to environmental stress relies on a wide range of tightly controlled regulatory mechanisms, including transcription. Changes in chromatin structure and organization accompany the transcriptional response to stress, and in some cases, can impart memory of stress exposure to subsequent generations through mechanisms of epigenetic inheritance. In the budding yeast *Saccharomyces cerevisiae*, histone post-translational modifications, and in particular histone methylation, have been shown to confer transcriptional memory of exposure to environmental stress conditions through mitotic divisions. Recent evidence from *Caenorhabditis elegans* also implicates histone methylation in transgenerational inheritance of stress responses, suggesting a more widely conserved role in epigenetic memory.

## 1. Introduction

Here we review recent studies showing how histone methylation contributes to the mitotic inheritance of transcriptional changes induced by stress in *S. cerevisiae*, and specific cases where exposure to high temperature, chemicals, and starvation have been associated with changes in histone methylation in *C. elegans*, a model system widely used for transgenerational studies. The examples described in *C. elegans* suggest that histone modifications may also play a role in transgenerational inheritance of environmental exposure, although most of the observed effects extend across a few generations at most, and the underlying mechanisms remain to be described. Nonetheless, the tractability of *C. elegans* continues to provide valuable insights into the complex and varied mechanisms of transgenerational inheritance.

Histone post-translational modifications (PTMs) are covalent modifications of histones that contribute to chromosome structure and function. In most species, repressive chromatin is generally enriched in H3K9 and H3K27 trimethylation (H3K9me3 and H3K27me3), while H3K4me3 is enriched at promoter regions and associated with more accessible chromatin [1]. In *S. cerevisiae* H3K9 methylation is absent, but other features of both repressive and active chromatin are conserved. In *C. elegans*, transcriptionally silent regions marked by H3K9me3/H3K27me3 and enriched in transposons and other repetitive sequences, are found predominantly on the arms of chromosomes, while the distribution of H3K4me3 is similar to that found in other organisms [2]. PTMs and their readers influence fundamental processes, including transcription, replication, and the maintenance of genome integrity. In a developmental context, regulation of the chromatin landscape is pivotal in the establishment and maintenance of the epigenetic programs of development and differentiation, and in the response to environmental stress [3].

## 2. Histone Methylation and Transcriptional Memory of Stress Induced Gene Activation in Yeast

Unicellular organisms such as yeast survive environmental stress by responding rapidly to changes in temperature, nutrients, and osmotic pressure, among others. In specific cases, survival is improved when yeast cells “remember” a previous stressful experience, and are capable not only of a more rapid reactivation of the appropriate transcriptional stress response, but also of transmitting this response to their descendants. Here, we focus on examples of epigenetic transcriptional memory implicating chromatin modifications, although alternative mechanisms have been reported [4].

### 2.1. INO1 Transcriptional Memory

One of the best-studied examples of epigenetic transcriptional memory has been described for yeast cells that have previously experienced inositol deprivation. Lack of inositol triggers the transcription of *INO1*, encoding inositol-1-phosphate synthase (the rate limiting enzyme for synthesis of inositol-containing phospholipids), and its relocalization to the nuclear periphery where it interacts with the Nuclear Pore Complex (NPC). *INO1* association with the NPC is mediated by the transcription factors Put3 and Cbf1 that bind two recruitment sequences (GRS I and GRS II) present in the *INO1* promoter [5,6]. Addition of inositol triggers transcriptional repression of *INO1*, but the gene remains anchored to the NPC through a different mechanism that relies on binding of the transcription factor Sfl1 to the Memory Recruitment Sequence (MRS_INO1_), and on interactions with different components of the NPC, e.g. Nup100 [7,8,9]. Persistent localization of *INO1* at the nuclear periphery results in faster transcriptional reactivation when inositol depletion reoccurs, and is inherited mitotically for up to four generations. Deposition of chromatin modifications at the *INO1* promoter is another component essential for memory. More specifically, incorporation of the histone variant H2A.Z and dimethylation of H3K4 (H3K4me2) are required for the ultimate output of memory: the recruitment of a poised RNA polymerase II (RNAPII) allowing faster transcriptional reactivation [8,9,10]. This final step also relies on the interaction of RNAPII with a specific form of the Mediator Complex containing the Cdk8 kinase [9].

Importantly, characterization of PTMs at the *INO1* promoter during the different transcriptional states has emphasized the pivotal role played by Set1/COMPASS, the only H3K4 methyltransferase present in yeast [11], in the regulation of transcriptional memory (Figure 1). During transcriptional repression, the *INO1* promoter is hypoacetylated and hypomethylated on H3K4. Upon activation, H3/H4 acetylation at the promoter is accompanied by Set1/COMPASS dependent H3K4 di- and tri-methylation [9]. Finally, as transcriptional memory is induced, histone H3 and H4 are deacetylated, and H3K4me2 prevails over H3K4me3. This shift between methylation states depends on the activity of a remodeled version of the Set1/COMPASS complex lacking the Spp1 subunit, which in yeast has H3K4 mono- and dimethylation activities, but is incapable of H3K4 trimethylation [12,13,14,15,16,17]. H3K4me2 enrichment at the *INO1* promoter requires both Sfl1 and MRS_INO1_, and is maintained during memory through an interaction with the histone deacetylase complex SET3C by mechanisms yet to be identified [9] (Figure 1). 

### 2.2. GAL Transcriptional Memory

Another well-characterized example of yeast transcriptional memory is activated by galactose, and applies to several inducible genes (*GAL1*, *GAL2*, *GAL7*, *GAL10*) required for galactose catabolism. The expression of *GAL* genes depends on the Gal4 transcription factor, which binds to the promoter of each gene regardless of the carbon source available. When galactose is absent, the transcriptional repressor Gal80 interacts with Gal4 and inhibits its transcriptional activity. Conversely, when galactose is present Gal80 is retained in the cytosol, allowing Gal4 to activate the *GAL* genes [18]. When naive yeast cells are switched from glucose to galactose medium, derepression of *GAL* genes occurs very slowly. However, when the same population is exposed to galactose for a second time after growing for several hours in repressive glucose medium, activation of the *GAL* genes occurs much faster. Memory of *GAL* transcription persists through up to seven cell divisions, the longest so far reported in *S. cerevisiae* [7]. Gal1, a regulatory protein capable of preventing Gal80 nuclear translocation, is instrumental for the establishment of memory. Accumulation of Gal1 during galactose exposure, followed by its cytoplasmic inheritance through mitosis upon glucose repression, allows faster removal of Gal80 inhibition and, consequently, faster *GAL* gene reactivation once cells re-experience galactose [19]. 

Interestingly, Gal1 is responsible for *GAL* gene targeting to the nuclear periphery both under activating conditions (galactose) and during *GAL* memory [7]. Similarly to what is observed for *INO1*, *GAL1* localization to the NPC during memory requires Nup100 and a specific MRS in the promoter (MRS_GAL1_). However, in contrast to the *INO1* memory mechanism, disruption of *GAL1* localization at the nuclear periphery does not affect *GAL1* transcriptional memory [20]. An additional regulator of *GAL* memory is Tup1 (both a transcriptional activator and repressor), which mediates the effect of Gal1 on *GAL1* peripheral localization as well as the binding of a poised RNAPII to the *GAL1* promoter. Importantly, Tup1 is also required for incorporation of H2A.Z and dimethylation of H3K4 at the *GAL1* promoter [20]. Similarly to what is observed for *INO1*, H2A.Z, and most likely H3K4me2, mediate the recruitment of RNAPII, which contributes to faster *GAL1* reactivation during memory. Therefore, both the accumulation of long-lived Gal1 and the chromatin-driven binding of RNAPII to the *GAL1* promoter are essential to establish memory.

### 2.3. Hormesis-Based Transcriptional Memory

Transcriptional memory is also associated with hormesis, an adaptive response to mild environmental stress through which the system improves its tolerance to more severe challenges [21]. Notably, yeast cells previously exposed to mild osmotic stress (0.7M NaCl) acquire both H_2_O_2 _ resistance and the ability to mount a faster transcriptional response to H_2_O_2_ treatment. Both effects last for up to four generations, although they depend on different mechanisms. H_2_O_2_ resistance is due to the accumulation of a stable catalase (Ctt1, a H_2_O_2_ -detoxifying enzyme) that, by analogy with Gal1, is slowly diluted through cell division in the post-salt-stress period. The second effect, faster response to H_2_O_2_, represents a form of transcriptional memory, the first reported to include a large set of genes, both activated and repressed [22]. Intriguingly, salt-stress memory requires Nup42 but not Nup100, suggesting that it may rely on an interaction with the NPC through a mechanism different from those described for *INO1* and the *GAL* genes [22]. Nonetheless, similarities have also been reported between salt-induced and other transcriptional memories. For instance, H3K4me2 and RNAPII are detected at the promoters of several genes induced by salt. Furthermore, a role for the SET3C deacetylase complex in maintaining H3K4me2 and promoting RNAPII recruitment appear to be conserved in both *INO1* and salt-stress memories [9]. Lastly, an enrichment of Cdk8 is also observed at the promoters of genes displaying salt-stress memory, suggesting that, as for *INO1* memory, the presence of Cdk8 on Mediator is required to activate transcriptional memory [9].

Strikingly, several aspects of yeast transcriptional memory are conserved in human cells. In HeLa cells, treatment with interferon-γ (IFN-γ) promotes transcriptional memory of a large number of genes that are more rapidly reactivated when cells are exposed to IFN-γ for a second time [23]. This memory is inherited mitotically for at least four generations and, as in yeast, is associated with H3K4me2 and RNAPII recruitment at the promoters of the affected genes. Moreover, Cdk8 is detected at these promoters, and Nup98 (orthologue of yeast Nup100) is essential for memory, although most likely through a distinct mechanism occurring in the cytoplasm rather than at the nuclear pore [8,9]. Taken together, evidence from yeast and mammalian cells suggests important roles for nuclear pore proteins and chromatin modifications, and in particular H3K4 methylation, in the establishment of transcriptional memory through the Cdk8+/Mediator-dependent recruitment of poised RNAPII. 

## 3. Transcriptional Memory of Repression in Yeast

The vast majority of examples of transcriptional memory in yeast concerns activation of genes. However, hormetic memory induced by osmotic stress also includes numerous repressed genes [22]. Recently, another example of transcriptional repression memory (TREM) has been reported in response to carbon source shifts [24]. More specifically, galactose-driven repression of hundreds of genes occurs more rapidly in if yeast cells previously exposed to galactose. Notably, TREM requires the histone deacetylase complex Rpd3L, which is targeted to the promoter of TREM genes by the interaction of its Pho23 subunit with H3K4me3. Based on these results, the authors propose that TREM occurs as a consequence of histone deacetylation targeted by H3K4me3, in turn provoking loss of RNAPII at the promoter of TREM genes [24]. Of note, in this particular setting promoter H3K4me3 plays a repressive role, as already described in other contexts [25]. Therefore, yeast Set1/COMPASS plays key roles in both memory of transcriptional activation and repression through its ability to generate either H3K4me2 or H3K4me3, depending on the presence of the Spp1 component.

## 4. Transgenerational Inheritance in *C. elegans*

Transgenerational memory in mammals has been mostly correlated with changes in DNA methylation, an established carrier of epigenetic information [26]. However, the causality of these effects has been challenged because of the global epigenetic reprogramming of primordial germ cells (PGCs) during early embryogenesis, and transgenerational inheritance in mammalian species remains an area of extensive debate (see references [27,28] for recent reviews on the subject). *C. elegans* lacks DNA methylation on cytosines [29], and recent evidence suggests that histone modifications may play an important role in the transmission of epigenetic information in this model organism. Data from several labs has shown that gene silencing initiated by exogenous double-stranded RNA (dsRNA) or piwi-interacting RNAs (piRNAs) can be stably inherited, with transgenerational effects lasting more than 20 generations [30]. Transmission of these small RNAs has also been implicated in imparting memory of the response to environmental stressors, including starvation and high temperature. However, in most of these cases memory of the inherited state is observed for only a few generations [31,32,33,34]. Here we will discuss recent examples of epigenetic inheritance of environmental exposure mediated by H3K9me3, H3K27me3 and H3K4me3. Importantly, it should be noted that in order to be considered transgenerational, and to exclude the possibility of effects due to direct exposure to the trigger, the trait observed should be transmitted at least to the F3 generation following exposure of P0 mothers, and the F2 generation following exposure of P0 males [35].

### 4.1. Repressive H3K9 and H3K27 Tri-Methylation and Transgenerational Transgene Desilencing

H3K9 and H3K27 tri-methylation have been implicated in heritability by exogenous dsRNA [30], although recent data suggest that their role may be limited to the establishment of heritable transgenerational silencing rather than its inheritance [36,37]. However, in several models of environmental stress under laboratory conditions, H3K9me3 was shown to mediate inheritance of phenotypes over several generations. The longest lasting effect was observed in animals carrying an integrated multicopy array consisting of GFP under control of the heat inducible *daf-21/Hsp90* promoter (*pdaf-21::GFP*). Following exposure to high temperature (25 °C), GFP expression from the array strongly increased, and 14 generations were required for it to return to basal levels when animals were transferred to 20 °C after five generations at 25 °C. Even more surprising, exposure to the higher temperature for a single generation was sufficient to impart memory for 7 generations following a shift to 20 °C (Figure 2A). This memory effect was not observed for a single copy transgene, suggesting that inheritance depends on the repetitive nature of the transgene [38]. 

Immunofluorescence experiments combined with DNA fluorescence in situ hybridization (DNA FISH) showed that at 20 °C, F2 embryos derived from P0 ancestors developed at 25° C accumulated less repressive H3K9me3 on the *pdaf-21::GFP* array than F2 embryos derived from control P0 developed at 16 °C, consistent with these arrays having features of heterochromatin that are sensitive to temperature (Figure 2A). No differences were observed in Polycomb mediated repressive H3K27me3, or in either H3K36me3 or H3K4me2, associated with active chromatin. Importantly, this difference was apparent in early embryos before the onset of zygotic transcription, and was inherited through both oocytes and sperm, indicating that the altered chromatin is not a secondary response to transcriptional changes in the embryo following heat shock. Although it is not known whether the histone methylation patterns themselves are responsible for transmitting memory of *daf-21*/*Hsp-90* expression, memory was found to be dependent on the SET-25 histone methyltransferase required for deposition of H3K9me3, while inactivation of small RNAs had no effect [38]. Transcription profiling showed increased expression of repetitive sequences in the absence of *set-25* up to three generations after a return of heat exposed animals to low temperature. These results suggest that changes in heterochromatin structure triggered by environmental stress may transmit epigenetic information between generations. How the increase in temperature leads to the loss of histone methylation marks deposited by SET-25 remains to be addressed. Interestingly, temperature-dependent phase separation of chromatin domains into liquid-like foci with physical properties critical for silencing has been described [39]. It is tempting to speculate that a related phenomenon may be implicated. 

Global changes in H3K9me3 were also found to be associated with transgenerational germline defects in progeny of animals exposed to Bisphenol A (BPA) [40], a widely used plastic chemical highly prevalent in human samples and associated with endocrine disruption [41]. The authors showed that a heterochromatic transgene array transcriptionally silenced in the germline was desilenced in animals exposed to BPA, and in their F2 and F3 progeny (Figure 2B). The authors also showed increased embryonic lethality and germline apoptosis in F3 progeny of exposed animals, consistent with compromised germline function (Figure 2B). Mating of F1 progeny derived from exposed P0 mothers with unexposed males did not rescue germline desilencing, indicating that the primary mode of inheritance of the BPA effect is through the female germline. Perdurance of the effects in the F3 generation is consistent with an epigenetic, rather than a maternally inherited effect. Chromatin immunoprecipitation sequencing (ChIP-seq) of whole adult worms at the F3 generation revealed a minor decrease in H3K27me3 for genes upregulated in F3 progeny of BPA treated mothers, as well as reductions in both H3K9me3 and H3K27me3 from the distal, largely heterochromatic chromosomal regions (Figure 2B). Immunofluorescence analysis suggested that at least some of this global reduction is observed in germ cells. This was supported by experiments showing that inactivation of the H3K9me3/H3K36me3 demethylases JMJD-2 and the H3K27me3 demethylase JMJD-3/UTX-1 re-established transgene silencing as well as H3K9me3 and H3K27me3 in F3 animals [40].

### 4.2. H3K4 Methylation in the Inheritance of Stress Responses and Life History Traits

The above examples of epigenetic memory of exposure to stress rely on largely artificial systems based on the expression of a repetitive transgene. Kishimoto et al. [42] showed that, as in yeast, effects induced by hormesis could be passed transgenerationally to descendants. Animals exposed to various stressors, including heavy metal (arsenite), hyperosmosis (NaCl) and fasting during embryonic development exhibited increased resistance to oxidative stress and proteotoxicity as adults, and increased resistance was transmitted to the subsequent F2 and F3 generations grown under unstressed conditions (Figure 2C). Exposure of only male parents to stressors during development resulted in increased oxidative stress resistance and lifespan extension in the F1 descendants, showing that epigenetic information could be transmitted through sperm for at least one generation.

Because H3K4 methylation and COMPASS complex components required for its deposition mediate transgenerational effects on longevity in *C. elegans* [43], the authors investigated the role of COMPASS in regulating the transgenerational hormetic response. Knockdown of COMPASS components *wdr-5.1*, *ash-2* and the *SET1* homologue *set-2* in the P0 generation had no effect on the increased stress resistance of the parent, but abrogated it in F1 descendants. This suggests that H3K4me3 is not required for stress-induced hormesis in the exposed animals, but is required for its inheritance at least in the F1 offspring. Knock-down of *wdr-5.1* in the germline, but not intestinal or neuronal cells of exposed parental animals or their F1 offspring, led to suppression of increased resistance in F1 descendants. However, no significant difference in global H3K4me3 levels between stressed and unstressed animals was observed, and whether H3K4 methylation is able to transmit information through the male germline was not investigated. Additional factors were also identified as contributors to transgenerational inheritance, including known mediators of the stress response such as the insulin-like growth factor (IGF) signaling effector DAF-16/FOXO, the heat-shock factor HSF-1, and the transcription factor SKN-1/Nrf in the parental somatic cells, an interesting example of germ-to-soma communication.

In a related study, exposing *C. elegans* to high glucose concentrations was shown to have deleterious consequences on the germline of the F1 and to a lesser extent F2 progeny of exposed animals, resulting in a small decrease in the number of progeny [44]. Decreased brood size in F1 progeny was accompanied by increased resistance to oxidative stress and conveyed protection against neurodegeneration. As in the previous example, this effect was shown to depend on the COMPASS components SET-2 and WDR-5, but H3K4 was unaltered in affected F1 progeny, so that an indirect effect could not be ruled out. Furthermore, effects on stress resistance were not inherited beyond the non-exposed F1-F2 generation, showing that they reflect maternal inheritance rather than a transgenerational inheritance mechanism. 

Recent evidence suggests that H3K4 methylation may also play a role in transgenerational memory of life history traits. *C. elegans* has a variety of developmental responses to nutrient availability, and two of the most well studied ones are dauer diapause and L1 developmental arrest, also known as L1 diapause [45,46]. Interestingly, examples of epigenetic inheritance have been observed in both [31,47,48]. In the absence of food, L1 larvae survive starvation up to two weeks by remaining arrested in the first larval stage, L1, and this is reversible upon feeding [49,50,51]. Jobson et al. [47] characterized the long-term phenotypic consequences of L1 larval arrest. They found that animals recovering from extended periods of starvation (8 days) had reduced growth and fertility compared to animals starved only 1 day. These animals were also more sensitive to subsequent starvation, suggesting that the experience of extended starvation does not protect against subsequent starvation. Curiously however, the F1, F2 and to a lesser extent F3 progeny of individuals most severely affected by extended L1 arrest were more resistant to heat, suggesting some form of epigenetic inheritance of stress resistance.

Subsequent work suggests that histone methylation may play a role in the transgenerational effect described above [48]. Activation of AMP-activated protein kinase (AMPK) in response to starvation is a conserved mechanism that facilitates metabolic adjustment [52]. In *C. elegans*, disruption of the genes encoding the two AMPK catalytic subunits, AAK-1 and *-2*, causes starved L1 larvae to die prematurely following prolonged starvation [53]. The authors found that when surviving post-L1 diapause mutant animals reached the adult stage, they showed both somatic and reproductive defects, including a reduced brood size. Remarkably, the reduced brood size was observed for up to 7 generations (Figure 2D). Levels of H3K4me3 were increased in the precursor germline cells (PGCs) of emergent post L1 diapause *aak-1/2^−/−^* mutant animals up to at least 6 generations (Figure 2D). RNAi knock down of the genes coding for the COMPASS subunits SET-2, ASH-2 or WDR-5.1, but also of *set-16*, encoding a MLL related H3K4 methyltransferase, partially suppressed both sterility and the reduced brood size in the F1 generation of post L1 diapause *aak-1/2^−/−^* mutant larvae. A *set-2* knock-out also reduced global H3K4me3 levels of post L1 diapause *aak-1/2^−/−^* mutant larvae (Figure 2D). Interestingly, SET-2 protein levels appeared to be slightly higher in the PGCs of *aak-1/2^−/−^* mutant larvae, suggesting that AMPK may influence H3K4 levels through regulation of SET-2 protein stability. The transgenerational signal responsible for this increase in H3K4me3 and the genetic consequences of increased H3K4me3 in starved *aak-1/2^−/−^* worms and their progeny remain unknown. However, recent results showing that SET-2 also plays a role in the inheritance of another AMPK-dependent response, namely the mitochondrial stress response [54,55], suggests the intriguing possibility that AMPK signaling and changes in H3K4me3 may be interconnected processes contributing to transgenerational inheritance.

Liu et al [55] showed that *C. elegans* can adapt to mitochondrial stress induced by electron transport chain (ETC) inhibitors, and this response depends on N6-methyladenine (6mA) and SET-2 (Figure 2E). Animals exposed to antimycin, an ETC complex III inhibitor, showed dose-dependent developmental delays. Exposure of P0 parents protected progeny from developmental delay defects up to the F4 generation (Figure 2E). Similar results were obtained with other mitochondrial inhibitors. Crossing antimycin treated males with untreated hermaphrodites resulted in F1 progeny with increased resistance compared to progeny from untreated males, showing that adaptation is not due to a maternal effect. The ATP content and oxygen consumption levels of these F1 animals, two parameters of mitochondrial function, were also comparable to those of the F1 progeny from untreated animals. 

Most importantly, reduced activity of SET-2, but not other methyltransferases tested, severely impaired transmission of mitochondrial stress adaptation. ChIP-seq analysis of antimycin treated animals showed a global increase in H3K4me3 on promoters of F1 progeny. Mitochondrial stress adaptation was also shown to depend on N6-methyldeoxyadenine (6 mA), and genetic analysis suggested that 6mA may act downstream of H3K4me3 in this response. The mechanism responsible for the increase in H3K4me3 following the induction of mitochondrial stress was not explored in this study, and it is not known whether this increase is maintained across generations. Nonetheless, it is intriguing that under starvation conditions, AMPK activation dampens germline H3K4 methylation to preserve germline function [48], while adaptation to mitochondrial stress is associated with a global increase of this same mark [55]. These results suggest the interesting possibility that in both cases AMPK activation impacts H3K4me3 methylation with different consequences on the germline and soma, possibly constituting an interesting example of germline to soma communication impacting inheritance of environmental exposure. 

## 5. Perspectives

The yeast *S. cerevisiae*, in which examples of transcriptional memory through mitosis are well established, continues to provide useful insight into the molecular mechanisms responsible for transcriptional inheritance of environmental stress responses. Histone post-translational modifications, and in particular H3K4 methylation, have emerged as primary actors in inheritance, and at least some features of this mitotic memory mechanism may be conserved in mammals. More controversial is whether transcriptional memory of exposure to environmental stress can be inherited across multiple generations through the germline. This has become an area of intense research with potential impacts on human health and implications in evolutionary biology. Key to this question is how acquired information can be inherited given the extensive reprogramming of the genome accompanying germline development. Work on model systems such as *C. elegans*, in which environmental conditions can be tightly controlled and robust assays can be used to monitor inheritance, will continue to provide useful insights in understanding how epigenetic marks are established and maintained transgenerationally, and how cellular metabolism impacts these processes. In this context, the development of techniques allowing chromatin profiling to be carried out specifically on purified germline tissue will allow a more accurate description of histone modification profiles across generations and contribute to the identification of the molecular signatures accompanying transgenerational memory.

## Figures and Tables

**Figure 1 cells-08-00339-f001:**
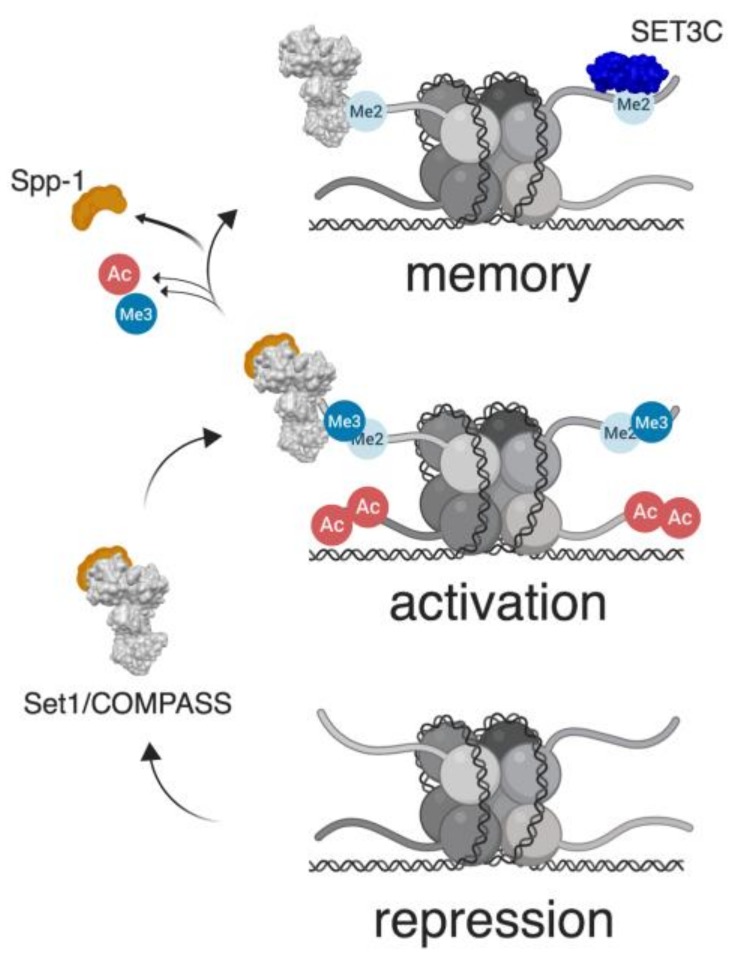
Model for *S. cerevisiae INO1* transcriptional memory. When *INO1* transcription is repressed (lower panel), nucleosomes associated with the *INO1* promoter are hypoacetylated and hypomethylated. Upon activation (middle panel), *INO1* relocalizes to the nuclear periphery, promoter nucleosomes are acetylated, and Set1/COMPASS di- and tri-methylates H3K4. When memory is induced (upper panel), *INO1* remains at the nuclear periphery, nucleosomes are deacetylated, and a remodeled version of Set1/COMPASS (lacking the Spp1 component required for H3K4 tri-methylation) leads to the accumulation of H3K4me2, which is maintained through its interaction with the SET3C histone deacetylase complex.

**Figure 2 cells-08-00339-f002:**
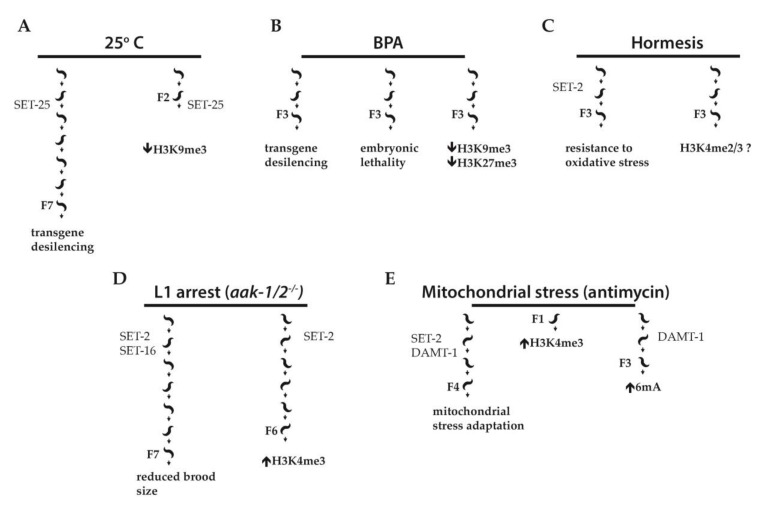
Examples of stress-induced transgenerational inheritance in *C. elegans*. (**A**) Worms exposed to high temperature (25 °C) for a single generation show desilencing of a multicopy *pdaf-21p::GFP* transgene for seven generations, dependent on the histone methyltransferase SET-25. F2 embryos derived from ancestors developed at 25 °C show decreased levels of H3K9me3 on the multicopy array as compared to controls developed at 16 °C. (**B**) A germline heterochromatic transgene array is desilenced by Bisphenol A (BPA). F3 progeny of treated worms show transgene desilencing, embryonic lethality, and a global reduction in H3K9me3 and H3K27me3. (**C**) Hormesis triggered by low doses of heavy metals, hyperosmosis, or fasting during embryonic development promotes resistance to oxidative stress in adult worms. The hormetic effect is transmitted to the F3 generation and, at least in the F1 generation, its presence requires the activity of SET-2. (**D**) In worms lacking AMPK activity (*aak-1/2^−/−^*), prolonged starvation-induced L1 arrest results in reduced brood size for up to seven generations. H3K4me3 levels are increased in PGCs of post L1 diapause *aak-1/2^−/−^* animals for up to at least six generations. SET-2 and the MLL related H3K4 methyltransferase SET-16 both contribute to the reduced brood size phenotype while SET-2 is also implicated in the post L1 diapause accumulation of H3K4me3 in *aak-1/2^−/−^* mutants. (**E**) Exposure of worms to the electron transport chain inhibitor antimycin induces developmental delay; antimycin treatment of parental worms protects future generations (up to F4) from developmental delay (mitochondrial stress adaptation). Loss of N6-methyldeoxyadenine methyltransferase DAMT-1 prevents 6mA accumulation. Both SET-2 and DAMT-1 are required for mitochondrial stress adaptation.

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
