# Peer review of "Histone Methylation and Memory of Environmental Stress"

_cells, 2019, doi:10.3390/cells8040339_

Round 1
Reviewer 1 Report
In this review, the author reviewed the histone methylation impart transcriptional memory of exposure to environmental stress in yeast and transgenerational inheritance in C.elegans. This review includes most of reference related this topic and well organized. There is some minor comments:
1 In abstract, they only mentioned the yeast but not emphasized the C. elegans; in addition, they should emphasize their discussion focus on these two models but not in mammalian cells.
2 In the first two parts, they introduced transcriptional memory in yeast while all the left context are in C. elegans. The third part is brief introduction of the later context. Therefore, it will be much easy to follow if the left context (from line 169) as a sub-par except perspective.
Author Response
Reviewer 1
In this review, the author reviewed the histone methylation impart transcriptional memory of exposure to environmental stress in yeast and transgenerational inheritance in C.elegans. This review includes most of reference related this topic and well organized. There is some minor comments:
We thank the Reviewer for his/her positive comments.
1 In abstract, they only mentioned the yeast but not emphasized the C. elegans; in addition, they should emphasize their discussion focus on these two models but not in mammalian cells.
We agree that C. elegans needs to be mentioned in the abstract. We modified this accordingly as shown hereafter:
Cellular adaptation to environmental stress relies on a wide range of tightly controlled regulatory mechanisms, including transcription. Changes in chromatin structure and organization accompany the transcriptional response to stress, and in some cases, can influence phenotypes in subsequent generations through what is collectively referred to as epigenetic inheritance. In yeast, histone post-translational modifications, and in particular histone methylation, can impart transcriptional memory of exposure to environmental stress conditions through mitotic divisions. Recent evidence from Caenorhabditis elegans also implicates histone methylation in transgenerational inheritance of stress responses, suggesting a more widely conserved role in epigenetic memory.
Concerning the last chapter, since this is a short Perspective section, we believe it is appropriate to mention very briefly how studies of epigenetic memory in simple model organisms might provide insights into memory in mammals.
2 In the first two parts, they introduced transcriptional memory in yeast while all the left context are in C. elegans. The third part is brief introduction of the later context. Therefore, it will be much easy to follow if the left context (from line 169) as a sub-par except perspective.
We agree with the Reviewer that the C. elegans part was not well organized. We added sub-sections as recommended. We also numbered each section.
Reviewer 2 Report
REVIEW REPORT
Review Manuscript
Title: “Histone methylation and memory of environmental stress”
by Fabrizio P.1, Garvis S.1 and Palladino F
This review manuscript is essentially addressing the link between epigenetics, particularly of specific histone modifications, and storage of information regarding life experience traits and environmental stress exposures. This review puts together key findings along the literature giving an important contribution by addressing a quite unexplored issue. The review is presented in a good structured manner.
A few specific suggestions include:
1- The title is very general. This review is particularly focused in yeast and the title should elucidate that.
2-Line 101: The “hormesis” concept is very interesting and should be emphasized, maybe by creating a specific section/title?.
3- Line 138- It’s not clear why only repression. Is there any examples of activation?
4- Line 151- “transgenerational inheritance in C. elegans” this should be completed mentioning what is inherited?
5- Lines 170-172. This paragraph is written in a confused manner. Could you clarify the message in this paragraph?
6- Line 183- “on the array” of what? Please, clarify.
7- Line 222- can you think in a way to introduce the controls in the scheme?
8- Line 240- Reformulate the title. Like it is, “and” is repeated twice.
9- Line 256-257- Please, in this sentence, clarify the connection between “stress-induced hormesis” and “offspring inheritance”.
10- Line 335- “inheritance of environmental exposure” please, specific and specify what do you mean and what is inherited?
Author Response
Reviewer 2
This review manuscript is essentially addressing the link between epigenetics, particularly of specific histone modifications, and storage of information regarding life experience traits and environmental stress exposures. This review puts together key findings along the literature giving an important contribution by addressing a quite unexplored issue. The review is presented in a good structured manner.
We thank the Reviewer for his/her positive comments.
A few specific suggestions include:
1- The title is very general. This review is particularly focused in yeast and the title should elucidate that.
We tend to disagree with the Reviewer on this point as the C. elegans part of the review is indeed slightly longer that the yeast part. Our intention was to give approximately the same space and importance to both models. Therefore, we think that the title is appropriate.
2-Line 101: The “hormesis” concept is very interesting and should be emphasized, maybe by creating a specific section/title?
We agree with the Reviewer on this point. We created individual sub-sections for each example of transcriptional memory including the hormesis-based memory.
3- Line 138- It’s not clear why only repression. Is there any examples of activation?
In the paper discussed (Lee BB et al, Nucleic Acids Res 2018, 46, 8261–8274) the authors found that approximately 1000 genes were differentially expressed in response to carbon shifts. They further showed examples of genes displaying memory of both activation and repression. However, the main focus of the paper was repression and the mechanisms behind it. To our knowledge, this is the first investigation of the mechanisms controlling memory of transcriptional repression and this is the part we wanted to emphasize in our review.
4- Line 151- “transgenerational inheritance in C. elegans” this should be completed mentioning what is inherited?
We understand that it would be appropriate to mention what is inherited. However, since this goes from transgene desilencing to brood size and mitochondrial stress adaptation, we felt it was not correct to give priority to any of those and we preferred a more general title.
5- Lines 170-172. This paragraph is written in a confused manner. Could you clarify the message in this paragraph?
We agree with the reviewer and have modified the text as follows:
H3K9 and H3K27 tri-methylation have been implicated in heritability by exogenous dsRNA [31], although recent data suggest that their role may be limited to the establishment of heritable transgenerational silencing rather than its inheritance [37,38]. However, in several models of environmental stress under laboratory conditions, H3K9me3 was shown to mediate inheritance of phenotypes over several generations.
6- Line 183- “on the array” of what? Please, clarify.
We added the name of the transgene, pdaf-21::GFP, which carries GFP under the control of the heat-inducible daf-21/Hsp90 promoter.
7- Line 222- can you think in a way to introduce the controls in the scheme?
We do not believe this to be necessary as this would significantly complicate the figure. However, controls for each experiment are detailed in the legend.
8- Line 240- Reformulate the title. Like it is, “and” is repeated twice.
We changed to title to: H3K4 methylation implication in the inheritance of stress response and life history traits.
9- Line 256-257- Please, in this sentence, clarify the connection between “stress-induced hormesis” and “offspring inheritance”.
We made the required change.
10- Line 335- “inheritance of environmental exposure” please, specific and specify what do you mean and what is inherited?
We modified the sentence to clarify what is inherited as follows:
The yeast S. cerevisiae, in which examples of transcriptional memory through mitosis are well established, continues to provide useful insight into the molecular mechanisms responsible for transcriptional inheritance of environmental stress responses.